# Homogenization Heat Treatment for Enhancing Corrosion Resistance and Tribological Properties of the Al5083-H111 Alloy

**DOI:** 10.3390/ma17133313

**Published:** 2024-07-04

**Authors:** Mohamed Balaid A. Rmadan, Ismail Esen, Hayrettin Ahlatci, Ece Duran

**Affiliations:** 1Mechanical Engineering Department, Karabuk University, Karabuk 78050, Turkey; mohamed.balaid@gmail.com; 2Metallurgical and Materials Engineering Department, Karabuk University, Karabuk 78050, Turkey; hahlatci@karabuk.edu.tr (H.A.); eceduran59@gmail.com (E.D.)

**Keywords:** Al5083-H111, Al-Mg, microstructure, immersion corrosion, wear, hydrogen evolution

## Abstract

In this study, an Al5083-H111 alloy was divided into two different parameters without heat treatment and by applying homogenization heat treatment. In the homogenized Al5083 sample, it helped to make the matrix structure more homogeneous and refined and distribute intermetallic phases, such as the Al-Mg phase (Mg_2_Al_3_) and Al-Fe phases, more evenly in the matrix. There was an increase in the hardness of the homogenized sample. The increase in hardness is due to the material having a more homogeneous structure. Corrosion tests were applied to these parameters in NaCl and NaOH. It is observed that Al5083 samples before and after heat treatment show better corrosion resistance and less weight loss in NaOH and NaCl environments. It was observed that the fracture resistance of the alloy in the NaOH solution was lower, and the weight loss was higher than the alloy in the NaCl solution. Wear tests were performed on two different parameters: a dry environment and a NaOH solution. Since the NaOH solution has a lubricating effect on the wear surface of the sample and increases the corrosion resistance of the oxide layers formed, the wear resistance of the alloys in dry environments was lower than the wear resistance of the alloys in the NaOH solution. A hydrogen evolution test was performed on the samples in the NaOH solution, and the results were recorded. Hydrogen production showed higher hydrogen output from the homogenized sample. Accordingly, a higher corrosion rate was observed.

## 1. Introduction

Al-Mg alloys are widely used in various fields due to their many properties, including their ability to change mechanical properties over a wide range, excellent formability in soft tempers, high resistance to corrosion, especially in marine environments, durability at both low and high temperatures, good weldability, lightness, and ballistic properties [1]. Among the many aluminum-based composites, Al 5083 is used worldwide due to its high specific strength and ease of manufacture. The Al5083 alloy, containing approximately 4.4% magnesium (5xxx series), is an excellent choice as a matrix for these composites due to its light weight, good ductility, weldability, corrosion resistance, and production ease, as well as its slightly lower density than usual [2]. Al 5083 alloys are important commercial alloys widely used in shipbuilding, offshore structures, and the transportation industries due to their high strength, good weldability, and exceptional corrosion resistance [3]. Since aluminum alloys are used in areas requiring corrosion and wear resistance, improving and investigating these properties in aluminum alloys is crucial. The excellent corrosion resistance property of the Al5083 alloy has been emphasized [4]. Mg is the main alloying element for Al 5xxx alloys, and the oversaturated Mg concentration in these alloys leads to the formation of the intergranular β phase (Al_3_Mg_2_) when exposed to high temperatures (50–200 °C) for a long time [5,6]. The β phase is anodic to the Al matrix and can easily corrode in saline environments [7,8]. The sensitivity to intergranular corrosion also depends on the Mg content, production, thermal process, corrosion solution, and the size of intergranular precipitates. Other alloying elements such as Mn, Cr, Fe, and Si are added to improve the mechanical properties [9]. The corrosion behavior of the Al5083 alloy in the NaCl solution has been a subject of interest in various studies. Seikh et al. [10] investigated the stress corrosion cracking behavior of the ECAP Al5083 alloy in a 3.5% NaCl solution using the slow strain rate tensile test. They found that the SCC behavior was influenced by the processing method of the alloy [10]. Elhasslouket al. [11] studied the effect of a 3.5% NaCl-10% HCl corrosive environment on the fatigue behavior of hot-rolled aluminum 5083-H111. Their findings indicated that corrosion in the NaCl solution led to the emergence of cracks originating from specific inclusions or corrosion pits, acting as stress concentrators [11]. Furthermore, Wang et al. [12] explored the corrosion behaviors of an as-rolled Mg-8Li alloy in differently concentrated NaCl solutions. They observed that the corrosion rate of the alloy in a 3.5 wt.% NaCl solution was significantly higher compared to a 0.9 wt.% NaCl solution after a certain immersion time. This suggests that the concentration of the NaCl solution plays a crucial role in the corrosion mechanisms of the alloy [12]. The corrosion behavior of the Al5083 alloy in the NaCl solution is affected by factors, such as the processing method of the alloy, the specific environment (e.g., NaCl concentration), and the presence of stress concentrators, such as inclusions or corrosion pits. The Al5083 alloy, known for its superior weldability, corrosion resistance, and impact resistance, is a popular structural material [13]. When exposed to a NaOH solution, the corrosion behavior of the Al5083 alloy is influenced by various factors. Studies have shown that the addition of certain elements can improve the corrosion resistance of aluminum alloys in alkaline solutions [14]. Furthermore, the mechanical properties and corrosion behavior of the Al5083 alloy can be enhanced through processes like friction stir processing, which can lead to grain refinement and improved corrosion resistance [15,16,17]. Research has also indicated that severe plastic deformation techniques, like equal-channel angular pressing (ECAP), can affect the corrosion behavior of the Al5083 alloy. ECAP has been shown to increase the strength and elongation of the alloy, leading to enhanced mechanical properties, which in turn can impact its corrosion resistance [10,18]. Additionally, the microstructure of the alloy plays a crucial role in its corrosion behavior. Studies have demonstrated that the microstructure of the Al5083 alloy can be modified through processes like friction stir welding, influencing its corrosion resistance [19,20]. The corrosion behavior of the Al5083 alloy in the NaOH solution is a complex interplay of factors such as alloy composition, microstructure, processing techniques, and the environment. Understanding how these factors interact is essential for improving the corrosion resistance of the Al5083 alloy in alkaline solutions. The degradation of the Al5083 and homogenized Al5083 alloys in NaCl can be affected by various factors such as microstructural properties, corrosion resistance, and mechanical properties.

Homogenization has been shown to effectively reduce shrinkage, porosity, and segregation in cast alloys [21]. Additionally, the microstructural evolution during homogenization processes can affect the corrosion resistance of alloys, like Al5083 [22]. Heat treatments play a crucial role in modifying the properties of aluminum alloys, such as Al5083. The study by Baghdadi et al. [15] focused on the mechanical property enhancement of Al5083/Al6061 joints through post-weld heat treatment (PWHT) and abnormal grain growth (AGG) control. This research highlighted the significance of PWHT methods in improving mechanical properties [15]. Al5083, being a non-heat treatable alloy, can benefit from heat treatments to enhance its mechanical performance. Sunnapu and Kolli [19] emphasized the impact of tool shoulder profile and rotational speed on the mechanical properties of friction stir welded Al5083 joints, indicating the importance of process parameters in achieving desired properties. Furthermore, the study by Tie et al. [23] demonstrated the effectiveness of heat treatment after the rheo-extrusion process in adjusting the mechanical and conductive properties of aluminum alloys. This suggests that heat treatment can be utilized to improve the mechanical and conductive properties of Al-Si-Mg alloys [23]. Wu et al. [24] showed that heat treatments significantly influence the microstructure of nickel–aluminum bronze alloys, consequently affecting their mechanical properties. This emphasizes the role of heat treatments in altering the microstructure and, consequently, the mechanical behavior of alloys [24]. Heat treatments have a profound impact on the microstructure and mechanical properties of aluminum alloys, like Al5083. By carefully selecting and applying appropriate heat treatment methods, it is possible to tailor the properties of these alloys to meet specific performance requirements. Homogenization heat treatment at 500 °C for 3 h is applied to the Al5083 alloy to eliminate segregation and tune the microstructure of the alloy ingot. This is a high-temperature heat treatment process consisting of heating and cooling steps performed at temperatures in the range of 450–600 °C [25]. This process aims to dissolve large-size eutectic phases, redistribute solutes, eliminate intragranular segregations, level out compositional variations, reduce internal stresses, and remove casting defects [26]. Homogenization heat treatment is crucial for traditional casting alloys, like Al5083, to improve their properties before subsequent processes, like hot extrusion [27]. The treatment helps in minimizing segregation and enhancing the performance of cast articles [28]. By subjecting the alloy to homogenization, the chemical composition segregation is reduced, and intermetallic formation at grain boundaries is altered, leading to a reduction in the corrosion rate [29]. The controlled cooling of samples during the homogenization heat treatment process significantly impacts the material’s microstructure and properties. Cooling in air, as opposed to quenching in water, leads to the formation of larger grains. Moreover, extending the duration of the heat treatment process results in increased grain sizes [30]. The rate of cooling post-homogenization is crucial in determining the final microstructure and properties of the material [31].

The use of Diamond-Like Carbon (DLC) and nanodiamond composite (NDC) hard coatings, deposited via Cathodic Arc Physical Deposition (CAPD), has shown promise in improving the corrosion and wear resistance of Al-Mg alloys [32]. Surface coatings, particularly DLC, are advantageous due to their low friction coefficient, amorphous structure, and excellent anti-corrosion and wear properties, making them well-suited for safeguarding Mg-based alloys [33]. Furthermore, the incorporation of nanodiamond particles in composite coatings has been shown to attract more nickel ions, leading to improved wear and corrosion resistance [34]. Studies have demonstrated that surface treatments, like plasma electrolytic oxidation, can effectively enhance the corrosion resistance of Al-Mg alloys, making them suitable for various applications, including in the aerospace industry and electronic components [35]. Additionally, the application of DLC coatings on Mg alloys has been found to significantly improve their surface performance and expand their utility in fields like aviation [36]. Moreover, the use of nanodiamond composite films has shown higher corrosion resistance and hardness, making them beneficial for enhancing the properties of coated materials [37].

Studies have shown that the electrochemical performance and discharge behavior of Al5083 alloys in NaCl solutions are affected by microstructure, with research conducted on cast, homogenized, and annealed conditions [38]. Corrosion is a significant concern for the performance and longevity of aluminum alloys, like Al5083. When exposed to aggressive environments, such as NaOH, the corrosion behavior of Al5083 and its alloys can be influenced by various factors. Pitting corrosion, which is the localized accelerated dissolution of the metal due to the breakdown of the protective passive film, is a common issue [39]. The corrosion behavior of Al5083 and its alloys in NaOH depends on factors such as microstructure, alloy composition, processing methods, and environmental conditions. Understanding these factors is crucial for developing strategies to enhance the corrosion resistance of these aluminum alloys in corrosive environments. Conflicting reports exist regarding the wear performance of aluminum alloys and aluminum-based composites at high temperatures. For instance, Al-Si-Mg (A356) alloys with and without SiC particles have been reported to exhibit mild to severe corrosion under dry sliding conditions due to frictional heating at approximately 0.4 Tm aluminum [2]. The microstructure of Al5083 greatly affects its wear performance. Processes such as friction stir processing have been found to impact the microstructure, mechanical properties, wear resistance, and corrosion behavior of Al5083 [40]. Studies also emphasize the importance of factors, such as particle size and distribution, in improving the wear behavior of aluminum composites [41]. Furthermore, the wear resistance of aluminum matrix composites can be attributed to factors like porosity elimination, homogeneity of particle distribution, and grain refinement [42]. In conclusion, the wear performance of Al5083 and its composites can be significantly enhanced by the addition of various reinforcements and processing techniques that improve the alloy’s mechanical properties and microstructure. Factors such as nanoparticle dispersion, particle size, and distribution play a crucial role in determining the wear resistance of these materials. The production of hydrogen using Al5083 and NaOH involves a process where aluminum reacts with sodium hydroxide to produce hydrogen gas. Hydrogen production through aluminum corrosion in a NaOH solution presents a cheaper and more economical method compared to hydrogen production via chemical hydride hydrolysis [43]. Among various alkaline solutions, the NaOH solution is the most preferred for hydrolysis reactions. When NaOH is the alkaline solvent, the reaction with aluminum is as follows [44]:2Al + 6H_2_O +2NaOH → 2NaAl (OH)_4_ + 3H_2_(1)
2NaAl (OH)_4_ → NaOH + Al (OH)_3_(2)
2Al + 6H_2_O → 2Al (OH)_3_ + 3H_2_(3)

NaOH is consumed in the hydrogen production reaction [1] and reappears through reaction [2]. The entire process can be summarized by reaction [3,45]. Consequently, only aluminum and NaOH are consumed, and hydrogen is produced [43]. The main obstacle to hydrogen production via this corrosion reaction is the easy passivation of the aluminum surface when Al is recovered with an Al(OH)_3_ layer [46]. When comparing the performance of KOH and Ca(OH)_2_ hydroxides used for hydrogen production, faster aluminum consumption was found in the NaOH solution [47]. Overall, the use of NaOH in hydrogen production processes involving aluminum, biomass, and other materials highlights its importance as a catalyst in promoting hydrogen production reactions. Its ability to facilitate reactions, increase reaction rates, and contribute to the formation of intermediates underscores its significance in the efficient production of hydrogen gas.

## 2. Materials and Methods

### 2.1. Material

In this study, the Al5083 -H111 alloy, whose content is given in Table 1, was used. For metallography, corrosion, wear, hydrogen evolution, and hardness tests, the aluminum plate was prepared by cutting parallel to the rolling direction (HYP).

In order to create two parameters from the samples and compare them, the Al5083 samples, cut in 10 × 10 × 10 dimensions, were wrapped in aluminum foil and embedded in SiO_2_ + graphite sand in the PLF Series 140–160 Protherm brand heat treatment furnace. They were heated at 500 °C for 3 h and then allowed to cool in air. Thus, tests were applied to the Al5083-H111 and Al5083-H111 homogenized samples.

### 2.2. Metallography

Two Al5083-H111 and Al5083-H111 homogenized samples, cut in 10 × 10 × 10 dimensions, were first sanded with 400, 600, 800, 1000, 1200, 2000, and 2500 grain SiC paper on a Mikrotest brand sanding and polishing device and polished on a polishing felt using 1 μm alumina paste. The polished samples were etched with a Keller separator (2 mL HF + 3 mL HCl + 5 mL HNO_3_ + 190 mL Water). A Carl Zeiss optical microscope was used for microstructural examination.

### 2.3. Hardness Test

Hardness measurements of the Al5083-H111 and Al5083-H111 homogenized samples were made with a BMS 3000-HB BRINELL brand hardness device. In the Brinell hardness test, the ball diameter was determined to be 5 mm, the load was 750 N, and the loading time was 10 s. The diameters of each sample were measured by creating 3 traces, and the hardness values were calculated by taking their average.

### 2.4. Wear Test

A forward–reverse wear tester was used for wear testing. The forward–reverse wear tester was carried out under constant load, constant speed, and constant distance. Wear tests were applied to the Al5083-H111 and Al5083-H111 homogenized samples in air and in a 5 M NaOH solution. The surfaces to be worn were sanded up to 1200 mesh and cleaned with alcohol. Wear tests were carried out under a 20 N load, a 0.1 m/s sliding speed, and a total sliding distance of 1000 m. The friction force during wear was measured by the load cell connected to the tribometer arm and instantly recorded on the computer. An AISI 52100 quality high-hardness steel ball was used as the abrasive tip material. The same procedures were repeated with new samples in the 5 M NaOH solution and under a sliding distance of 600 m.

### 2.5. Immersion Corrosion Test

For testing, the surfaces of the samples were sanded and cleaned with an ultrasonic cleaner. The surface area of each sample was measured individually, and the weight measurements were made using a Precisa brand precision scale. The NaCl and NaOH solutions were used for the test. For the immersion corrosion test in the NaCl and NaOH solution, first, the jars were placed in a 3.5% NaCl solution and the samples were immersed in these jars. Testing was recorded at 24, 48, and 72 h. The oxide layer formed by the sample molecules was cleaned at recorded hour intervals. For this process, first, the samples were cleaned using an ultrasonic cleaner in a chromic acid solution dissolved in 180 g/L of distilled water. Then, pure water was used to clean the chromic acids in the sample and, finally, the sample was cleaned with ethanol and dried. After drying, the weight measurements of the samples were subjected to repeat intervals. As a result of the immersion rate test, the decrease in weight change and corrosion rates of the sample in certain hours were calculated. The same procedures were repeated using the new sample in a 5 M NaOH solution for 48 h.

### 2.6. Hydrogen Evolution Test

For the experiment, a 5 M NaOH solution was added to the glass beakers and covered to prevent air escape. A tourniquet tube was connected to the arm of the beaker, and the other end of the tube was connected to the tip of the glass syringe. The experiment was started after the Al5083 sample was placed in the NaOH solution. Glass syringes with a 100 mL capacity and a 2 mL sensitivity step used in Dissolved Gas Analysis were used in the hydrogen evaluation test. When hydrogen evaluation begins, the pressure pushes the syringes backward. When the experiment was started, time was kept, and the experiment was terminated when the syringe reached 100 mL.

## 3. Results and Discussion

### 3.1. XRD Patterns

XRD results of the Al5083-H111 alloy are given in Figure 1. While the Al_3_Mg_2_, Al_6_Mn, Al_12_Mg_17_, and Mg_2_Si phases were seen in the majority of the XRD standard cards, the AlMn phase was also seen in small amounts. XRD peaks of the Al5083 alloy started at 37.5° with the Mg_2_Si, Al_6_Mn, Al_12_Mg_17_, and Al_3_Mg_2_ phases. The second peak occurred at 43.6° with the Mg_2_Si and Al_6_Mn phases. The highest peak occurred at 37.5° degrees as the initial peak. XRD peaks of the Al5083 alloy ended with the Al_6_Mn and Al_3_Mg_2_ phases at 81.48°. Kılınç et al. [48] identified the presence of the Al_3_Mg_2_, Mg_2_Si, and Al_6_ (Mn-Fe-Cr) phases in the AA5083 alloy. In contrast, Kurnaz et al. [49] reported the formation of the brittle Al_12_Mg_l7_ phase in the grain boundaries of Al-Mg alloys.

### 3.2. Microstructure

Optical microscope images of the original and homogenized states of the Al5083 alloy are given in Figure 2 and Figure 3 respectively. The light-colored regions in the image largely represent the main matrix phase, namely, α-Aluminum (Al) phase. This phase forms the basic structure of the Al5083 alloy and largely determines its mechanical properties. The dark-colored regions and dots generally represent intermetallic compounds present in the alloy. The Al-Mg phase (Mg_2_Al_3_) and Al-Fe intermetallic phases are observed. These phases are usually observed as dark lines or dots in the microstructure. During the cooling and solidification of the alloy, the second phases formed by the insoluble components in the solution were determined to be small particles dispersed in the matrix. The rolling direction can be seen in the microstructure images. In the homogenized Al5083 sample, it helped to make the matrix structure more homogeneous and refined and to distribute intermetallic phases, such as the Al-Mg phase (Mg_2_Al_3_) and Al-Fe phases, more evenly in the matrix. The homogenization process made the grain boundaries more distinct and regular and reduced the grain structure. It was observed that dislocations and agglomerations decreased.

SEM micrographs and EDS analyses of the H111 hot-rolled Al5083 alloy before the heat treatment and after homogenization heat treatment are shown. The dark-colored structure at point 1 in Figure 4 and Table 2 is thought to consist of the Al_12_Mg_17_ and Al_3_Mg_2_ phases. Intermetallic phases were observed dispersed within the grain boundary and grain. It is thought that the AlMn phase is also seen in addition to the Al_12_Mg_17_ and Al_3_Mg_2_ phases at point 2, and the presence of a small amount of Fe was seen in the EDS analysis. It is thought that the gray contrast structure inside the black circle at point 1 in Figure 5 and Table 3 consists of the Mg_2_Si, Al_3_Mg_2_, and Al_6_Mn phases. It is thought that the reason for the high Si level in the EDS analysis is due to the homogenization heat treatment. It is thought that the dark structure at point 2 consists of the Al_12_Mg_17_, Al_3_Mg_2_, and AlMn phases. A more even distribution is observed during homogenization compared to before heat treatment.

### 3.3. Hardness Test Results

Brinell hardness (HB) measurements of the Al5083 alloys were made and compared before and after the heat treatment. The hardness value (Figure 6), which was 52.97 HB before the heat treatment, increased to 69.62 HB after the homogenization process. There was an increase in the hardness of the homogenized sample. The increase in hardness is due to the material having a more homogeneous structure. It was thought that brittleness increased with the increase in hardness. Homogenization heat treatment leads to increased hardness in materials for several fundamental reasons. First, homogenization processes can lead to a reduction in the size of structural elements within the material, such as carbide particles in alloys. This reduction in size results in an increase in stiffness, as smaller structural elements are associated with higher strength properties [50]. Additionally, homogenization processes can promote the formation of a more uniform microstructure throughout the material [51,52]. This uniformity helps minimize microstructural differences, leading to a more homogeneous structure and, therefore, increased hardness [50].

### 3.4. Wear Test Results

The changes in the weight of the H111 hot-rolled Al5083 alloy before the heat treatment, after homogenization, in dry environments, and in the NaOH solution as a function of distance are shown in Figure 7. The wear rates at the end of 10,000 m are comparatively given in Figure 8, and the friction coefficients measured during wear are presented in Figure 9. After dry environment wear tests, the Al5083 sample before the heat treatment exhibited greater weight loss, while the homogenized sample showed better wear resistance. The wear test results of the investigated samples in NaOH exhibit similarities. This can be explained by the more homogeneous and stable crystal structure and the increase in hardness value. Comparing the wear results with the hardness results, it is seen that the hardness results support the wear results. When examining the wear results in NaOH, it is determined that the homogenized sample shows better wear resistance, parallel to the results in a dry environment. In the wear test conducted in NaOH, lower wear resistance is expected compared to the dry environment. However, as seen in Figure 7 and Figure 8, higher wear resistance and weight loss are detected in the wear results in NaOH.

Since homogenization heat treatment is a critical process that significantly affects the microstructure of alloys [53,54], in the H111 hot-rolled Al5083 alloy, the homogenization heat treatment eliminated segregation and refined the microstructure of the alloy ingot. This treatment is aimed at achieving a more uniform distribution of elements throughout the material by reducing segregation. In this case, it can profoundly affect the mechanical properties [15,54,55] and lead to a reduction in corrosion rates [56]. In summary, the homogenization heat treatment of the H111 hot-rolled Al5083 alloy results in a refined and more uniform microstructure by eliminating segregation and tuning the distribution of elements throughout the material. This treatment leads to improved mechanical properties and corrosion resistance, underscoring the significance of homogenization processes in enhancing the performance of the alloy.

The higher wear resistance of the Al5083 and homogenized Al5083 samples in the NaOH solution compared to the dry environment can be attributed to several factors. It is thought to be due to the lubricating effect of the NaOH solution. The presence of the NaOH solution alters the chemical environment around the aluminum surface, leading to the formation of a protective oxide layer that acts as a barrier against further corrosion. This protective oxide layer is crucial in reducing the corrosion rate of aluminum in alkaline solutions, like NaOH. Additionally, the concentration of OH^−^ ions in the NaOH solution plays a role in the corrosion behavior of aluminum; higher OH^−^ ion concentrations can enhance the passivation of the aluminum surface, further reducing the corrosion rate [57]. Moreover, the interaction between the aluminum surface and the NaOH solution affects the corrosion process. Studies have shown that with increasing NaOH concentration, the corrosion potential shifts to more negative values, indicating a change in the corrosion behavior of aluminum in alkaline solutions [58]. The presence of Na^+^ ions in the solution can also influence the corrosion behavior of aluminum, as they can participate in the formation of corrosion products on the aluminum surface [59]. It was determined that the alloy exposed to wear in the NaOH solution after homogenization provided the least weight loss and the best wear resistance after 10,000 m. The weight loss and wear rate at 6000 m for this material are 1.09 × 10^−2^ g and 5.45 × 10^−8^ g/Nm, respectively. The highest weight loss and worst wear resistance were observed for the alloy. The weight loss and wear rates at the end of 10,000 m are 3.23 × 10^−2^ g and 16.18 × 10^−8^ g/Nm, respectively. According to these results, as shown in Figure 9, the friction coefficients during wear are parallel, and the homogenized material has the lowest friction coefficient in the NaOH solution after homogenization, with a friction coefficient of 8.06 × 10^−3^.

SEM and EDS analyses of the worn surfaces of the H111 hot-rolled Al5083 alloy in a dry environment and NaOH before and after heat treatment and homogenization are shown in Figure 10. When the SEM image in Figure 10 is examined, in the gray area at point 1 (Table 4), faint wear marks and small amounts of broken pieces due to wear are seen clinging to the surface. At point 2, it is seen that there are deep wear marks and pieces broken off from the surface, and these pieces are stuck to the surface. There is a wave appearance due to the broken piece. At point 3, it is seen that the pieces that broke off from the surface during wear stick to the surface again, accumulate, and swell, creating a flake appearance. When the SEM image in Figure 11 is examined, at point 1 (Table 5), a slight wear mark is visible in the gray area, but no broken pieces sticking to the surface can be seen. At point 2, it was observed that the broken and broken pieces accumulated on the matrix surface in the gray contrast area, resulting in a wavy image. At the third point, it is seen that a larger piece is stuck to the matrix compared to the broken pieces in the light gray region. In general, fewer wear marks, deformation, and material loss were detected on the surface of the homogenized material compared to the unheat-treated sample. When the SEM image in Figure 12 is examined, faint wear marks are visible at point 1 (Table 6). On the other hand, there are no parts breaking off and no parts sticking to the matrix. At point 2, wear marks are seen in the gray-white contrast area. However, an oxide film layer is observed. At point 3, wear marks in the light gray wavy area and a small number of broken pieces after wear are seen adhering to the matrix. When the SEM image is examined in Figure 13, at point 1 in Table 7, it was observed that an oxide film layer formed in the area where light gray accumulation was observed. At point 2, it is seen that a small number of broken pieces on the surface are stuck to the matrix. At point 3, slight wear marks can be seen in the gray area. It was determined that homogenized Al5083 had lighter wear, less depth of wear marks, and less amount of breakaway wear compared to untreated Al5083. When the wear in a dry environment and a NaOH solution were compared, it was determined that wear in NaOH increased the wear resistance due to the lubricating effect of NaOH and the formation of an oxide film layer by reacting with Al5083.

### 3.5. Immersion Tests

In Figure 14, the change in weight loss of the Al5083 samples before the heat treatment and after homogenization in the NaCl solution after 72 h is given, and Figure 15 shows corrosion rates after 72 h. In Figure 16, the change in weight loss of the Al5083 samples before the heat treatment and after homogenization in the NaOH solution after 48 h is given, and corrosion rates after 48 h are shown in Figure 17. In both environments, the homogenized samples show better corrosion resistance and less weight loss. Homogenization plays a crucial role in influencing the corrosion behavior of various materials. Studies have shown that machining techniques, such as high-pressure torsion (HPT), can increase the corrosion resistance of materials, such as pure Mg, leading to a more homogeneous corrosion surface [60]. Additionally, a homogeneous distribution of corrosion inhibitors, such as graphene nanoplatelets, can significantly increase the long-term corrosion resistance of materials, such as aluminum [61]. Conversely, the lack of a corrosion barrier effect due to homogenization can reduce the corrosion performance of some alloys [62]. Homogenization can also affect stress corrosion cracking resistance; grain refinement and microstructure homogenization increase this resistance in some cases [63]. It has also been reported that the formation of a more homogeneous passive film due to homogenization increases the corrosion resistance of aluminum alloys [64]. Similarly, hot working processes that result in finer grains and a more homogeneous microstructure have been shown to reduce corrosion rates of magnesium alloys [65]. The homogenization process can lead to a reduction in corrosion rates by reducing chemical composition segregation and altering intermetallic formation [29]. Additionally, improvement and homogenization of microstructures have been found to increase the corrosion resistance of certain alloys by promoting the formation of protective passive films [66]. Homogenized microstructures can greatly affect the recrystallization behavior, grain size, mechanical properties, and corrosion resistance [67]. In summary, homogenization plays a vital role in changing the microstructure of materials, affecting the formation of passive films and affecting the distribution of corrosion inhibitors; all of which have been shown to contribute to the overall corrosion resistance of various alloys and composites.

It is seen that the weight loss in the NaOH solution is higher than in NaCl, and it shows lower corrosion resistance. Kharel et al. [68] found that alloys exhibited higher corrosion rates in NaOH solutions compared to NaCl, supporting the idea that NaOH promotes corrosion to a greater extent. Consequently, the higher corrosiveness of NaOH compared to NaCl can be attributed to accelerated corrosion processes in NaOH solutions, the dissolution of protective films, and specific chemical reactions occurring in the presence of NaOH. While the highest weight loss and corrosion rate were seen in the untreated Al5083 in the NaOH solution, the lowest weight loss was seen in the homogenized Al5083 sample in NaCl.

In the XRD card of the corrosion of the untreated Al5083 sample in the NaCl solution in Figure 18, the MgO, MgO + Al_2_O_3_, Al_2_O_3_, and SiO_2_ phases were seen at high rates. XRD peaks of the Al5083 alloy started with the MgO, MgO + Al_2_O_3_, Al_2_O_3_, and SiO_2_ phases at 37.3°. The highest peak is the second peak and occurred at 43.18° degrees. The MgO, MgO + Al_2_O_3_, and SiO_2_ phases were observed at the highest peak. XRD peaks ended with the MgO phases at 81.42°. In Figure 19, the MgO and SiO_2_ phases were seen at high levels on the XRD card of the corrosion of the Al5083 homogenized sample in the NaCl solution. XRD peaks of the Al5083 alloy started with the MgO, MgO + Al_2_O_3_, and Al_2_O_3_ phases at 37.84°. The second peak occurred at 44° with the MgO, MgO + Al_2_O_3_, and SiO_2_ phases and is the highest peak. The XRD peaks of the Al5083 alloy ended with the MgO phase at 81.62°. The XRD measurement of the corroded surface of both the untreated and homogenized Al5083 alloy in the NaCl solution reveals the formation of identical oxides in both samples. The enhanced corrosion resistance of the homogenized Al5083 alloy can be attributed to the formation of a more protective and uniform oxide layer on its surface, which can be achieved by the finer and uniformly distributed intermetallics in the alloy structure.

When the SEM images of the untreated sample after corrosion in the NaCl solution given in Figure 20 are examined, it is thought that there is fragment attachment after corrosion in the black-gray contrast structure at point 1 (Table 8). The dark structure at point 2 is thought to be the formation of a residual oxide film. At point 3, the structure seen with gray contrast is seen to have corrosion in the form of flaking, while there are deposits around the flaking. While it is thought that there is material loss due to corrosion, it is also seen that an oxide layer forms on the surface. When the post-corrosion SEM images of the homogenized sample in the NaCl solution given in Figure 21 are examined, the white shiny structure at point 1 (Table 9) is thought to be MgO + Al_2_O_3_. The gray-colored structure at point 2 is thought to be Al_2_O_3_. The faint gray structure at point 3 is thought to be pitting corrosion and is thought to be a SiO_2_ oxide layer. There appears to be pitting corrosion in the matrix, Al_2_O_3_, and MgO. It has been observed that oxide layers form and accumulate in large structures within the areas pitted by this corrosion. The corrosion rate was enhanced in the non-heat-treated sample due to the existence of fragment detachment, oxide film residue, and flaking corrosion damage. However, the corrosion resistance was improved by the creation of narrow pittings with a homogeneous oxide film.

XRD results of the untreated and homogenized Al5083-H111 alloys after corrosion in the NaOH solution are shown in Figure 22 and Figure 23. XRD peaks of the Al5083 alloy started with the MgO, NaOH, and MgO + Al_2_O_3_ phases at 37.82°. All phases were seen at the highest peak. MgO was observed in high density in all peaks. XRD peaks of the Al5083 alloy ended with the MgO phase at 81. 66°. In the XRD card of the wear test in homogenized Al5083 NaOH, the MgO, SiO_2_, and NaOH phases were seen at high levels in Figure 23. XRD peaks of the Al5083 alloy started with the MgO, NaOH, MgO + Al_2_O_3_, and SiO_2_ phases at 38°. The second peak occurred at 44.04° with the MgO, NaOH, SiO_2_, and MgO + Al_2_O_3_ phases and is the highest peak. The XRD peaks of the Al5083 alloy ended with the MgO phase at 81.7°.

In X-ray diffraction (XRD) analysis of the untreated and homogenized Al5083-H111 alloys corroded in NaOH, the presence of a high MgO + Al_2_O_3_ phase can have important consequences in various materials science applications. The combination of MgO and Al_2_O_3_ can lead to the formation of spinel phases, such as MgAl_2_O_4_, which have been detected in different studies. These spinel phases are known to exhibit specific diffraction patterns in XRD analysis, indicating their presence in the material. The formation of spinel phases, such as MgAl_2_O_4_, is affected by the composition of the materials. The presence of MgO and Al_2_O_3_ is very important for their development [69,70]. Additionally, the ratio of MgO to Al_2_O_3_ can affect the phase composition of materials. For example, a high MgO content favors the production of certain phases, such as pyroxene, while a high Al_2_O_3_ content is more conducive to the production of other phases, such as anorthite [71]. This highlights the importance of understanding the MgO/Al_2_O_3_ ratio in material synthesis to control phase formation. Moreover, the diffusion of Mg_2_+ from MgO to Al_2_O_3_ can lead to the formation of a spinel network, such as MgAl_2_O_4_, which is very important in various applications [72]. The interaction between MgO and Al_2_O_3_ through diffusion processes plays an important role in the formation of spinel phases and affects material properties. In summary, the presence of a high MgO + Al_2_O_3_ phase detected by XRD analysis indicates the formation of spinel phases, such as MgAl_2_O_4_, which is affected by the MgO/Al_2_O_3_ ratio and diffusion processes between MgO and Al_2_O_3_. Understanding these aspects is essential for tailoring material properties in various fields of materials science. Prabhakar et al. [16,73,74] found that the grain refinement and the homogeneous and fine dispersion of intermetallics within the structure associated with the friction stir process can increase the corrosion resistance of the Al5083 alloy. A number of studies have demonstrated the effect of various coatings formed during corrosion testing on the corrosion resistance of Al alloys. Ryu and Hong [75] found that the KF-NaAlO_2_ electrolyte resulted in a thick MgAl_2_O_4_ coating, which exhibited the highest corrosion potential and polarization resistance. Ardelean et al. [76] observed a shift in the corrosion potential and a decrease in the anodic dissolution current in magnesium and its alloys with cerium, aluminum oxide, and aluminum hydroxide surface films. Kameneva et al. [77] highlighted the influence of TiN, ZrN, and TixZr1-xN layers on the corrosion resistance of a hard alloy in a sodium hydroxide solution, attributing the differences to surface and internal defects of the coating layers. These studies collectively support the role of Al_2_O_3_, MgO, and SiO_2_, films in slowing down the corrosion process by reducing the charge transfer rate and diffusion flow through the surface layer (such as chloride ion diffusion) [78]. The presence of phases of elements such as Al, Mg, and Si that form the oxide film increases corrosion resistance through mechanisms, such as the inhibition of oxychlorination [16,70,72,79]. These studies show that the incorporation of MgO, MgO + Al_2_O_3_, and SiO_2_ oxide films can play a crucial role in reducing corrosion in the Al5083 alloy. In addition, studies by Jha and Bhattarai [80] on sputter-deposited W-xNb alloys showed that corrosion rates in solutions containing different proportions of NaOH were higher than those in 3.5% NaCl. Hussein [78] highlighted the positive effect of coatings containing SiO_2_ nanoparticles in improving corrosion resistance. Revealing the positive effect of SiO_2_ added to composite coatings in the form of nanoparticles [78] contributes to the understanding of increasing corrosion resistance by the SiO2 film formed on the surface of metal alloys, such as Al5083. In the Al5083 sample that was not treated in the NaOH solution, unlike the homogenized sample, the SiO_2_, MgO + Al_2_O_3_, and Al_2_O_3_ phases were not seen. The absence of these phases reduces corrosion resistance. The increase in corrosion resistance of the homogenized sample is directly proportional to the increase in oxide films. 

As shown in Figure 8, the lower weight loss in the corrosion tests performed in the NaOH solution compared to dry environments can be attributed to the accumulation of alkali on the surface and the formation of thin oxide films, as observed from the corrosion XRD analysis results (Figure 22 and Figure 23). This situation is supported by the lower friction coefficient obtained in the wear tests in the NaOH solution, as shown in Figure 9.

When the post-corrosion SEM images of the unheat-treated sample in the NaOH solution given in Figure 24 are examined, the gray structure in the pit at point 1 (Table 10) is thought to be the NaOH oxide layer. The small structure at point 2 is thought to be a MgO and Al_2_O_3_ oxide film. It is thought that the structure seen in the form of a white crust at point 3 is the accumulation of the NaOH oxide layer. It is observed that pitting corrosion occurs on the surface with the appearance of a seashell due to the solution effect. When the SEM images of the homogenized sample after corrosion in the NaOH solution given in Figure 25 are examined, it is thought that MgO and a small number of NaOH oxide layers are formed in the large white structure at points 1 and 2 (Table 11). It is thought that MgO + Al_2_O_3_ is formed in the gray structure seen as the dot. At point 3, the structure in the dark region is thought to be MgO and SiO_2_. It was observed that the seashell appearance on the surface decreased, and higher levels of oxide layers accumulated on the surface compared to the unheat-treated sample.

The corrosion of aluminum in various solutions, including NaCl and NaOH, has been extensively researched to comprehend its impact on the material’s behavior [68,80,81,82]. Alameer [82] has also delved into the corrosion behavior of Al matrix composites in NaCl, HCl, and NaOH solutions at different temperatures, revealing that the type of solution and temperature conditions significantly affect the corrosion behavior of Al alloys. The corrosion behavior of Al matrix composites in a 10% solution of sodium chloride (NaCl), sodium hydroxide (NaOH), and hydrochloric acid (HCl) was examined; the highest corrosion rate was in the HCl solution, which is excessively corrosive for most materials [82]. The Al alloy has been found to exhibit poor corrosion resistance when exposed to alkalis, such as the NaOH solution, as Al alloys are rapidly attacked by even dilute alkali solutions. Alameer [82] showed that Al alloy matrix composites exhibited a lower corrosion rate in the NaCl solution compared to other solutions, and also the corrosion rate increased with rising exposure time. This is due to the fact that chloride undergoes decomposition when exposed to water for an extended period of time. When examining the corrosion SEM images of Al5083 alloys in the NaCl solution (Figure 20 and Figure 21) and NaOH solution (Figure 24 and Figure 25), it was found that the corrosion surface in the NaOH solution had a rougher texture, and a clearly visible oxide film in the form of a layer in the NaCl solution was not observed in the NaOH solution. 

### 3.6. Hydrogen Evolution

Figure 26 shows the hydrogen evolution rate of the unheated and homogenized samples in the NaOH solution, and Figure 27 presents the corrosion rate graphs of these samples. In Figure 26, it is observed that the hydrogen output of the homogenized sample, although it starts slower, gradually increases and reaches the 100 mL hydrogen gas output level in a shorter time. The untreated sample’s gas evolution starts rapidly, then decreases, and subsequently shows a stable increase. When the corrosion rates are compared, there is an initial high increase followed by a decrease, and then a continuous increase is observed in the untreated sample. The homogenized sample exhibits a slow corrosion rate followed by a steady state.

When comparing the two graphs, it can be seen that hydrogen output and the corrosion rate are directly proportional. As the release of hydrogen gas increases, the corrosion rate also increases. The homogenization heat treatment has significant effects on the release of hydrogen gas and promotes gas formation. One reason for this is that the phases within the grain become coarser with homogenization and are distributed homogeneously into the matrix (Figure 24). Hydrogen gas production increases as the phases within the grain react with the NaOH solution. Homogenously dispersed phases increase the surface area and, consequently, the production of hydrogen gas. Research has shown that heat treatment can indeed affect hydrogen formation rates. Studies on the microstructural evolution of Al-5083, particularly the β phase (Al_3_Mg_2_) at grain boundaries and intragranular particles, indicate that the presence and distribution of phases can affect the hydrogen evolution behavior of the alloy. Goswami et al. demonstrated that the β phase at the grain boundaries of samples subjected to longer aging dissolved more during corrosion [83]. In the case of Al5083, the homogenization heat treatment may influence the hydrogen evolution behavior. Yao et al. found that the hydrogen evolution rate increased slightly after heat treatment, indicating a potential compromise in the post-treatment protection of the material [84]. Additionally, Petroyiannis et al. [85] showed that heat treatment at certain temperatures can release hydrogen trapped in corroded alloys, thereby affecting the properties of the material.

Hydrogen evolution is a significant phenomenon closely linked to corrosion processes in various materials, particularly metals like aluminum, magnesium, and zinc. The evolution of hydrogen during corrosion is a result of cathodic reactions that occur concurrently with anodic dissolution, influencing the overall corrosion rate [86]. Studies have shown that hydrogen evolution can take different forms, such as large stable bubbles on uncorroded regions, fine bubbles at the corrosion front, and medium-sized bubbles behind the corrosion front [87]. 

Furthermore, the connection between hydrogen evolution and corrosion is evident in the context of galvanic coupling, where higher rates of hydrogen evolution are observed on dark corroded surfaces compared to uncorroded surfaces. This increased hydrogen evolution catalyzes self-corrosion processes, leading to accelerated material degradation [88]. The advancement of the hydrogen front alongside the corrosion front in materials, like aluminum alloy 2024, demonstrates how hydrogen can penetrate materials through intergranular paths generated during the corrosion process [89]. 

Hydrogen evolution is intricately linked to corrosion processes in various materials, influencing corrosion rates, morphology, and localized corrosion behavior. Understanding the mechanisms of hydrogen evolution during corrosion is essential for developing effective corrosion mitigation strategies and improving the performance and durability of materials exposed to corrosive environments.

## 4. Conclusions

While the Al5083 sample showed more weight loss before heat treatment, better wear resistance was observed in the homogenized sample. When the wear results are compared with the hardness results, it is seen that the hardness results support the wear results. Considering the wear results carried out in NaOH, it was determined that the homogenized sample showed better wear resistance, parallel to that in the dry environment. Higher wear resistance and weight loss were detected in the wear results in NaOH. Lower corrosion and wear resistance of the Al5083 and homogenized Al5083 samples were observed in the NaOH solution compared to a dry environment. It is observed that the Al5083 samples before and after the heat treatment show better corrosion resistance and less weight loss in the NaOH and NaCl environment. It was observed that the NaOH solution had lower corrosion resistance and higher weight loss compared to the NaCl solution. In hydrogen production, higher hydrogen output from the homogenized sample and a correspondingly higher corrosion rate were observed.

## Figures and Tables

**Figure 1 materials-17-03313-f001:**
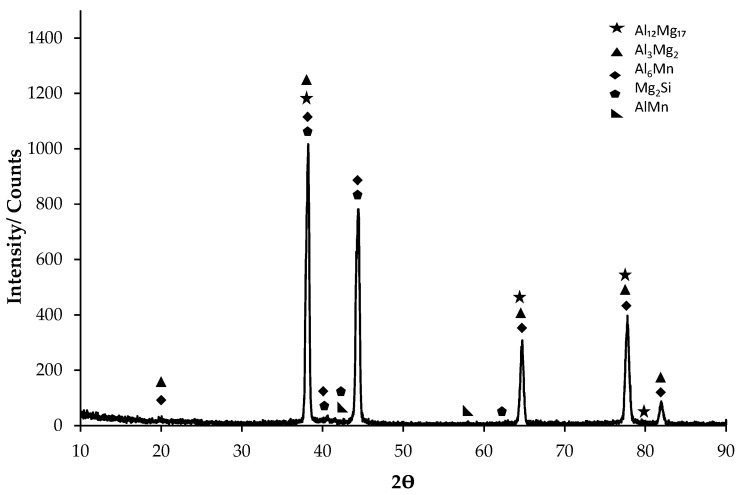
XRD patterns of the Al5083 alloy.

**Figure 2 materials-17-03313-f002:**
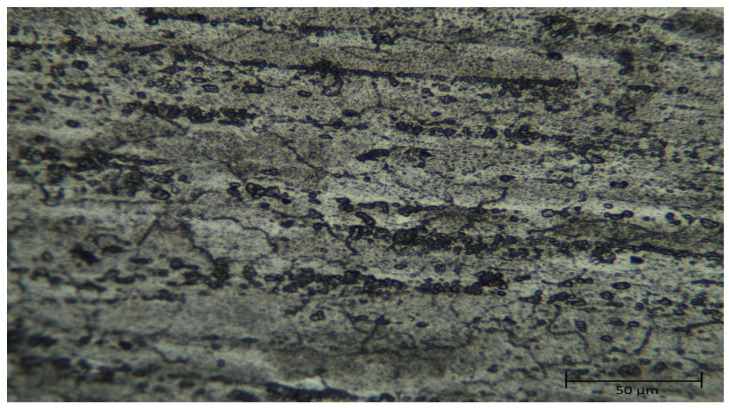
Optical microscope image of Al5083.

**Figure 3 materials-17-03313-f003:**
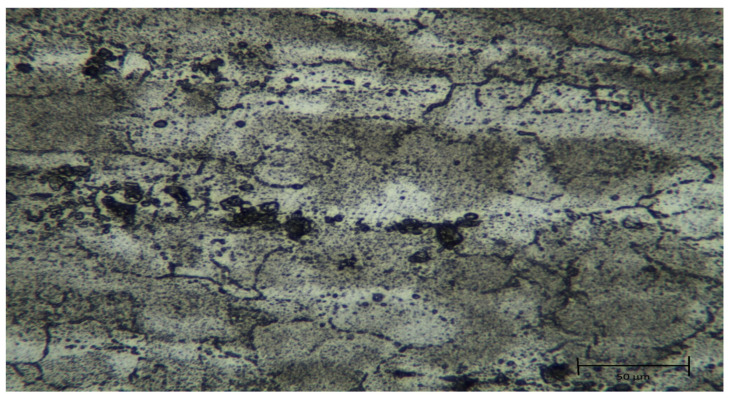
Optical microscope image of homogenized Al5083.

**Figure 4 materials-17-03313-f004:**
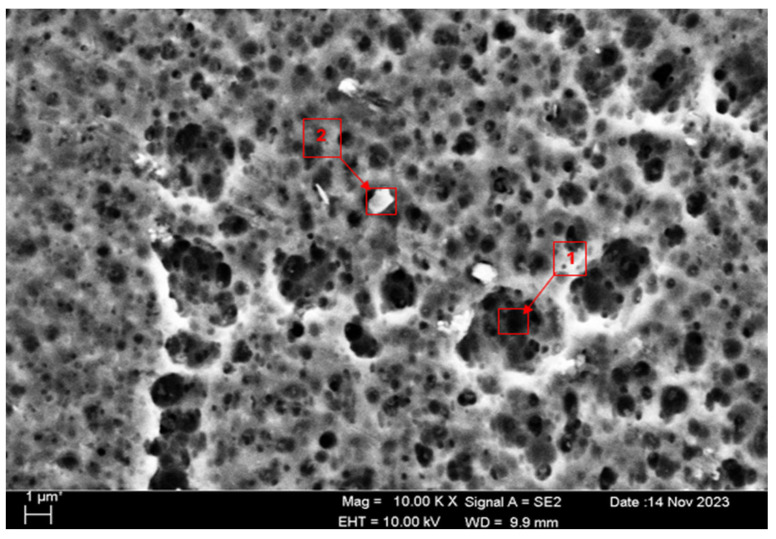
SEM micrograph of Al5083.

**Figure 5 materials-17-03313-f005:**
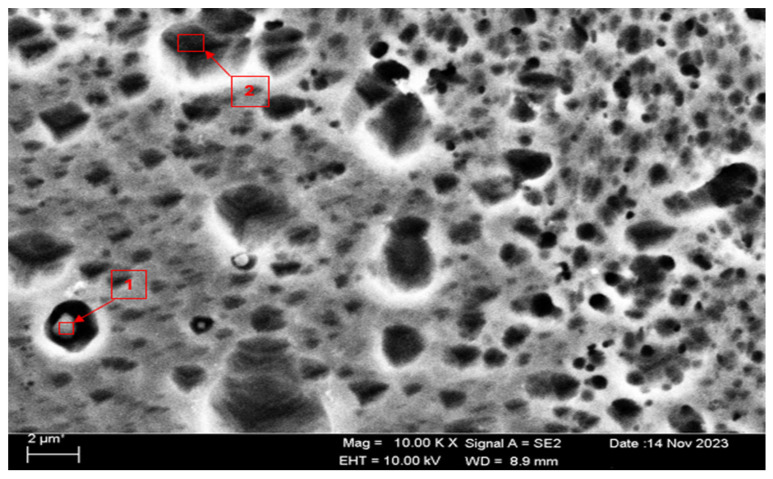
SEM micrograph of homogenized Al5083.

**Figure 6 materials-17-03313-f006:**
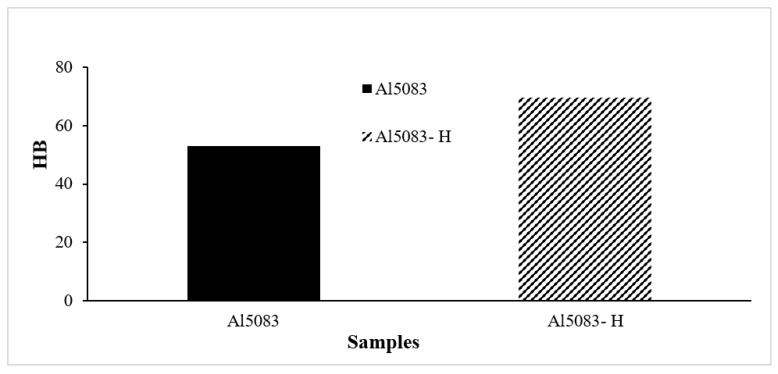
Hardness results of Al5083 and after homogenization (-H) of the Al5083 alloys.

**Figure 7 materials-17-03313-f007:**
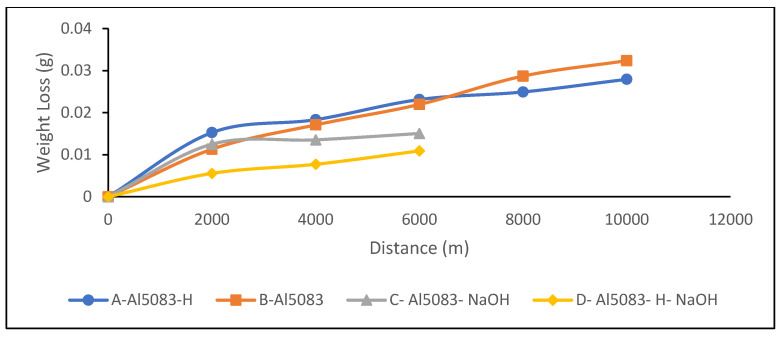
Weight loss of Al5083 and homogenized (-H) Al5083 alloys in dry media and NaOH.

**Figure 8 materials-17-03313-f008:**
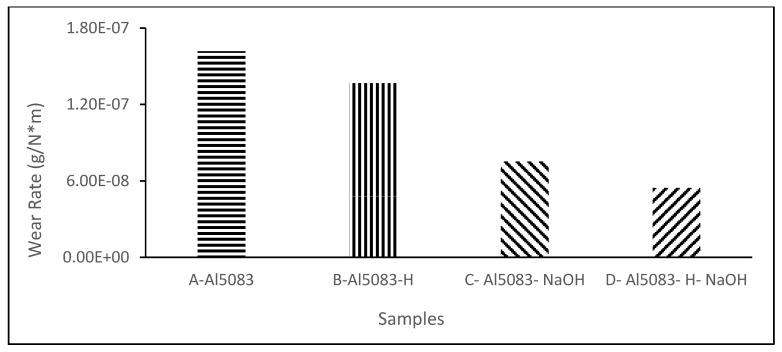
Wear rate of the Al5083 and homogenized (-H) Al5083 alloys in dry media and NaOH.

**Figure 9 materials-17-03313-f009:**
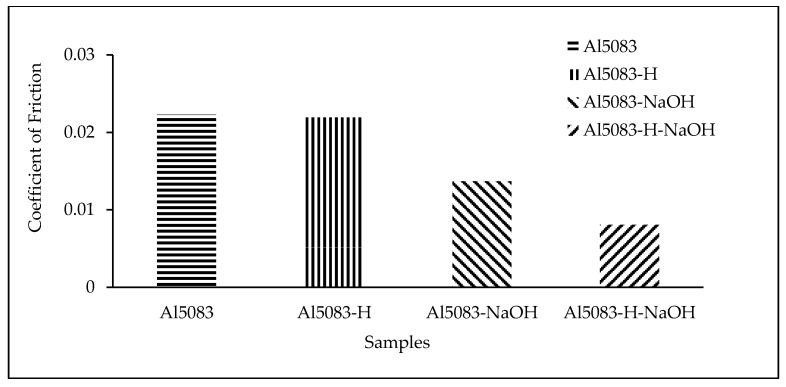
Coefficient of friction of the Al5083 and homogenized (H) Al5083 alloys in dry media and NaOH.

**Figure 10 materials-17-03313-f010:**
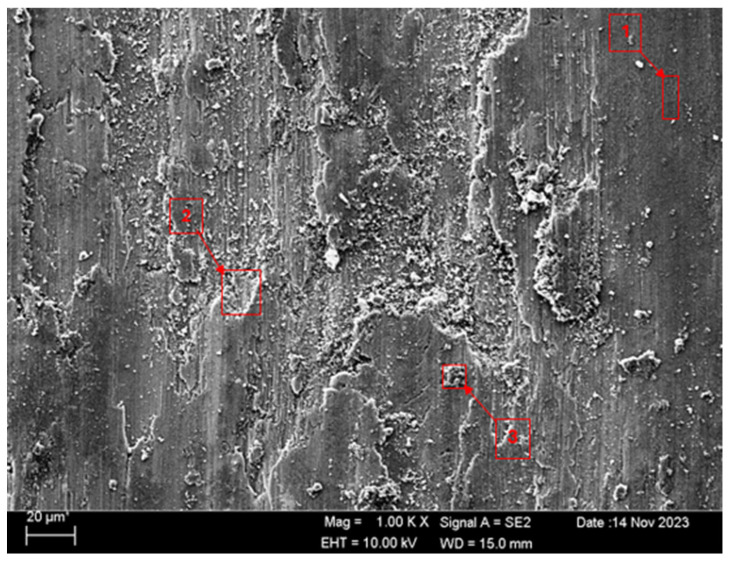
SEM image of Al5083 after a wear test in a dry environment.

**Figure 11 materials-17-03313-f011:**
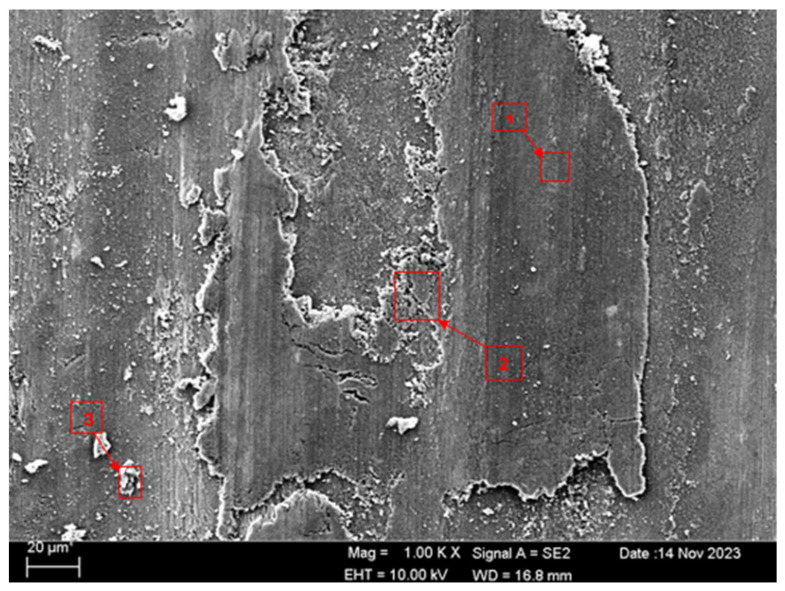
SEM image of homogenized Al5083 after a wear test in a dry environment.

**Figure 12 materials-17-03313-f012:**
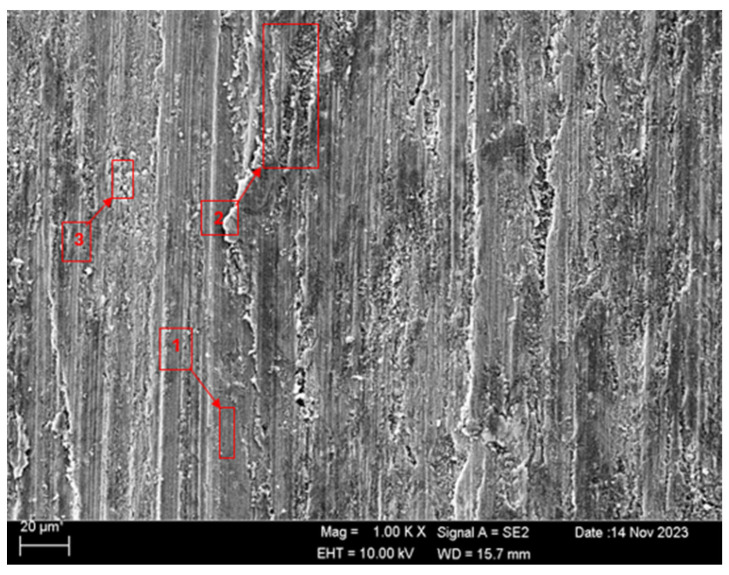
SEM image of Al5083 after a wear test in the NaOH solution.

**Figure 13 materials-17-03313-f013:**
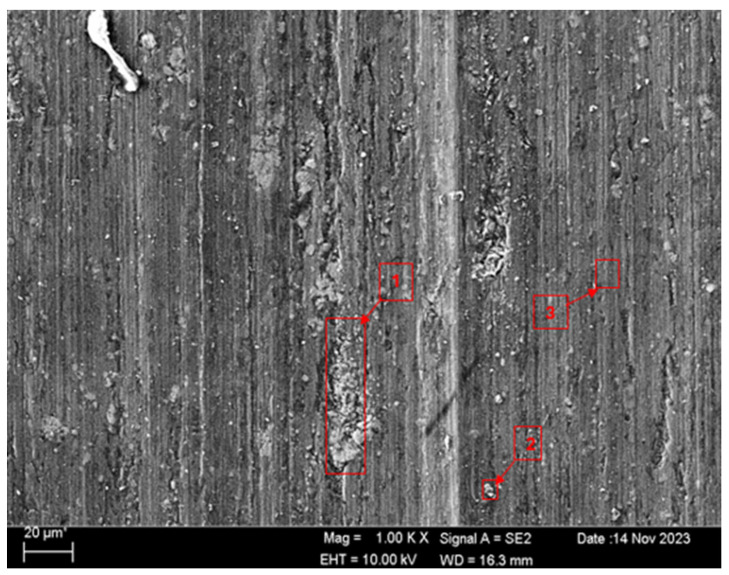
SEM image of homogenized Al5083 after a wear test in the NaOH solution.

**Figure 14 materials-17-03313-f014:**
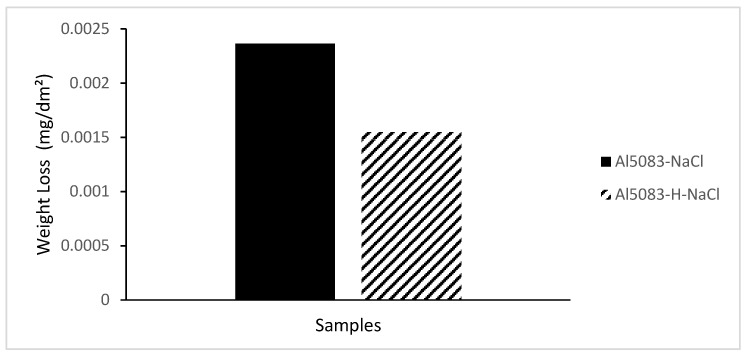
Weight loss after 72 h corrosion of Al5083 and homogenized (H) Al5083 in NaCl.

**Figure 15 materials-17-03313-f015:**
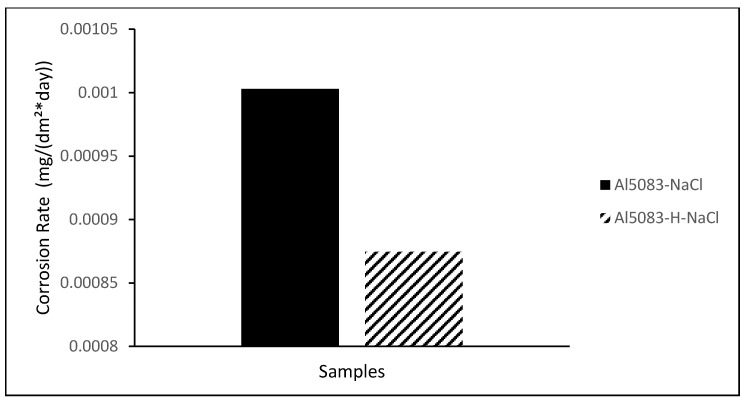
Corrosion rate after 72 h corrosion of Al5083 and homogenized (H) Al5083 in NaCl.

**Figure 16 materials-17-03313-f016:**
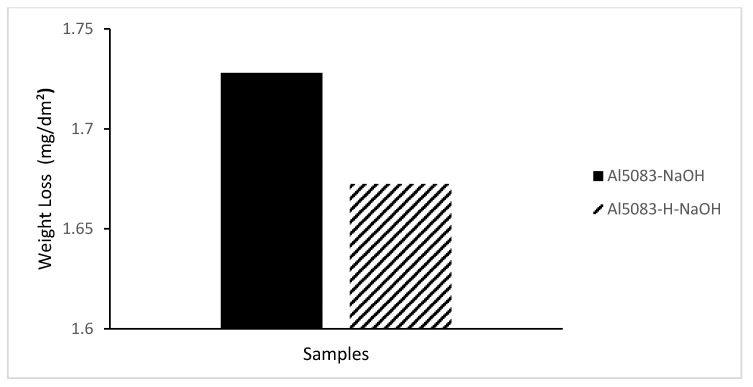
Weight loss after 48 h corrosion of Al5083 and homogenized (H) Al5083 in NaOH.

**Figure 17 materials-17-03313-f017:**
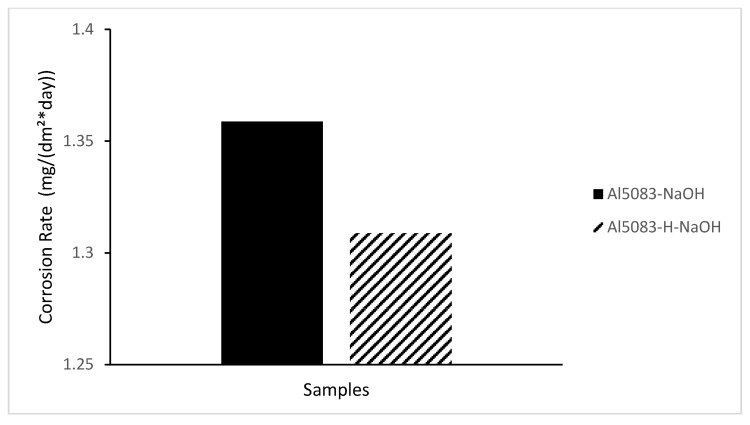
Corrosion rate after 48 h corrosion of Al5083 and homogenized (H) Al5083 in NaOH.

**Figure 18 materials-17-03313-f018:**
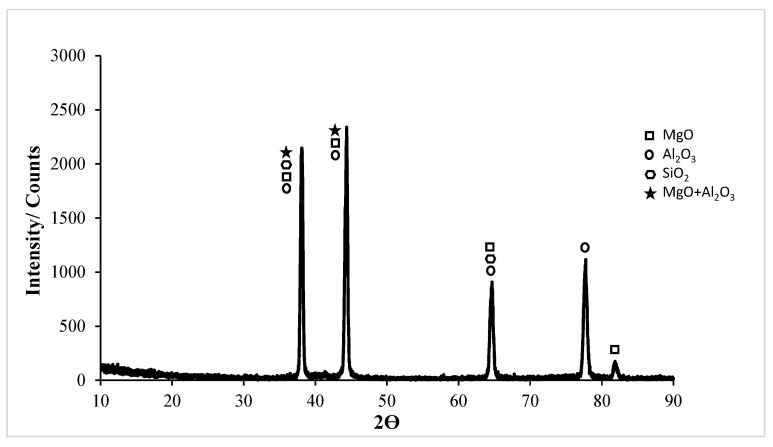
Corrosion XRD results of Al5083 in NaCl.

**Figure 19 materials-17-03313-f019:**
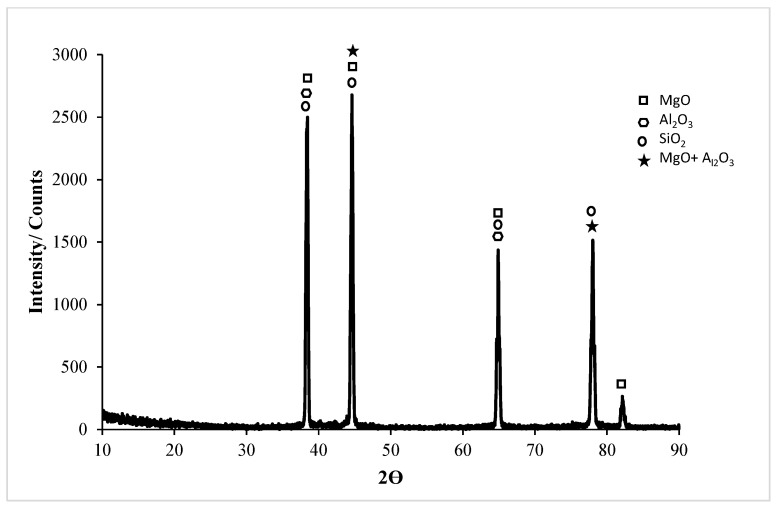
Corrosion XRD results of homogenized Al5083 in NaCl.

**Figure 20 materials-17-03313-f020:**
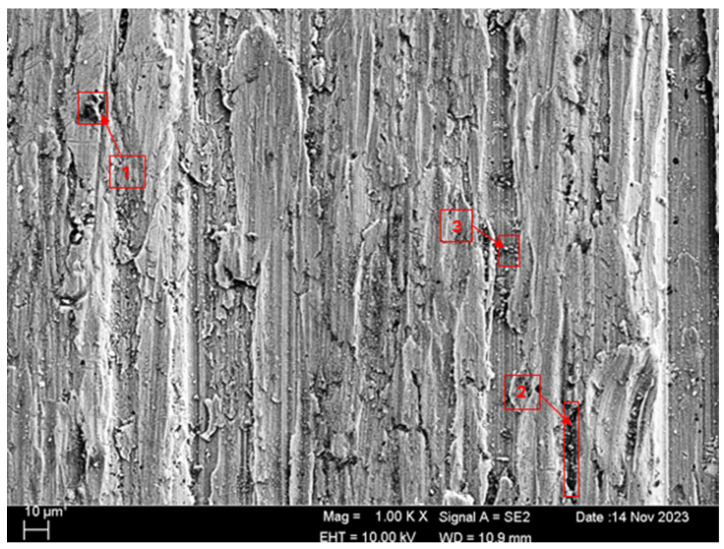
SEM image of Al5083 corroded in NaCl.

**Figure 21 materials-17-03313-f021:**
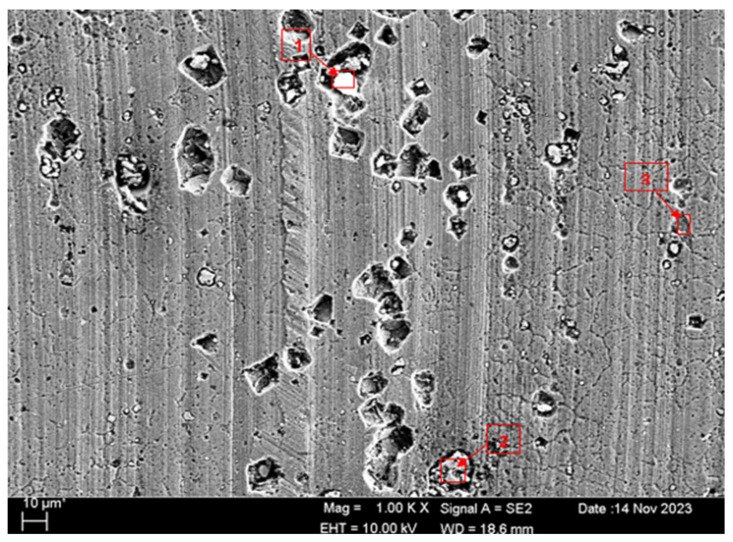
SEM image of homogenized Al5083 corroded in NaCl.

**Figure 22 materials-17-03313-f022:**
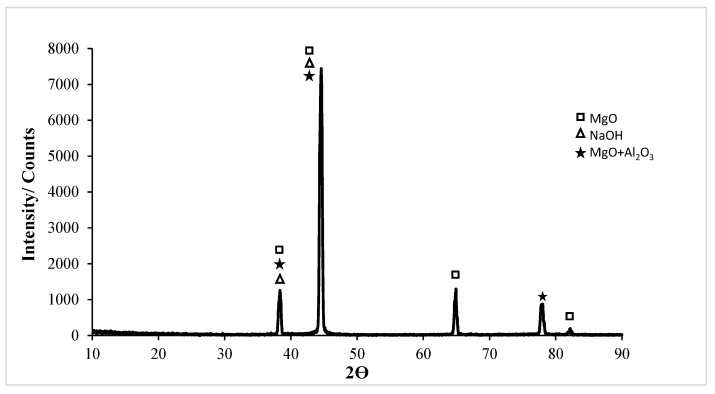
Corrosion XRD results of Al5083 in NaOH.

**Figure 23 materials-17-03313-f023:**
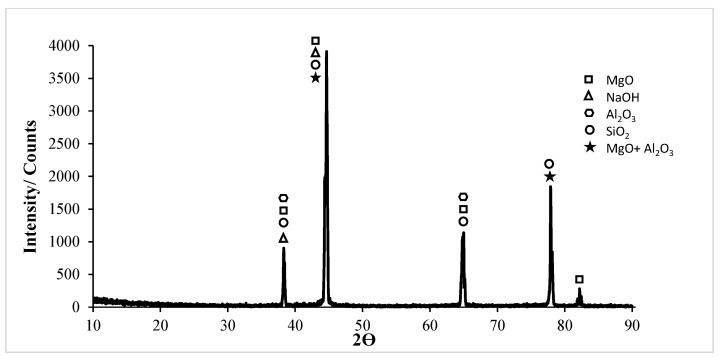
Corrosion XRD results of homogenized Al5083 in NaOH.

**Figure 24 materials-17-03313-f024:**
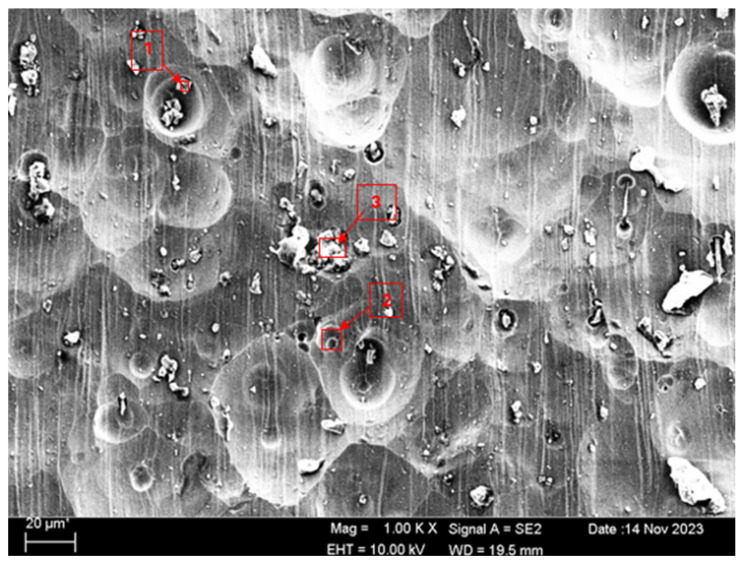
SEM image of Al5083 corroded in NaOH.

**Figure 25 materials-17-03313-f025:**
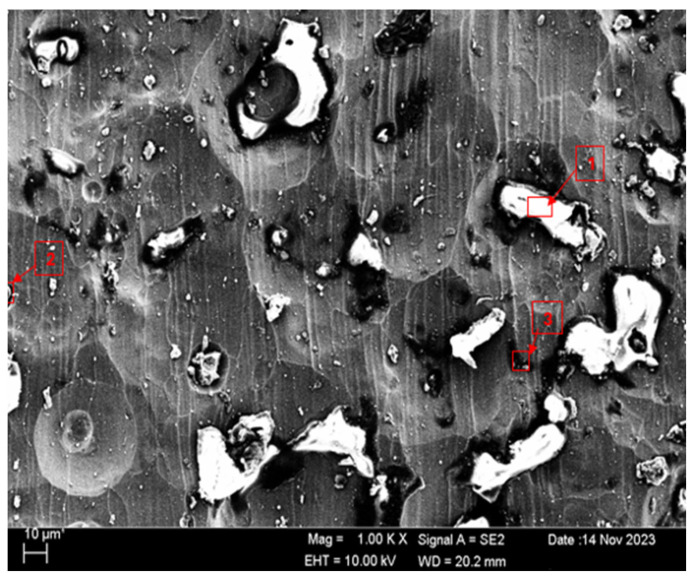
SEM image of homogenized Al5083 corroded in NaOH.

**Figure 26 materials-17-03313-f026:**
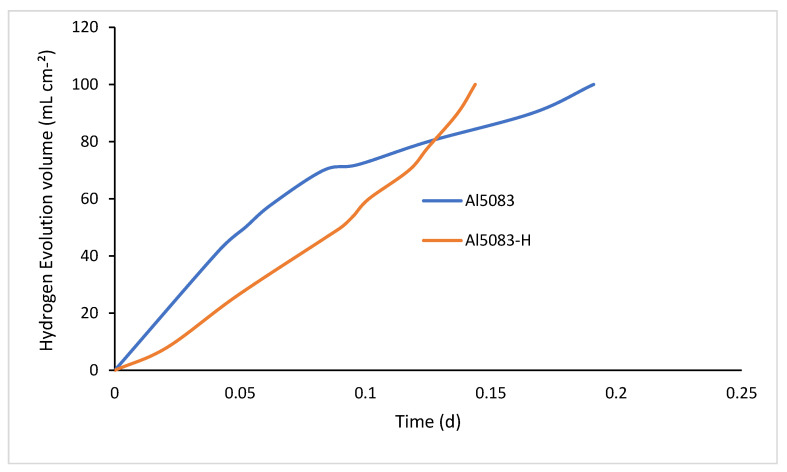
Hydrogen evolution volume results of Al5083 and homogenized (H) Al5083 in NaOH.

**Figure 27 materials-17-03313-f027:**
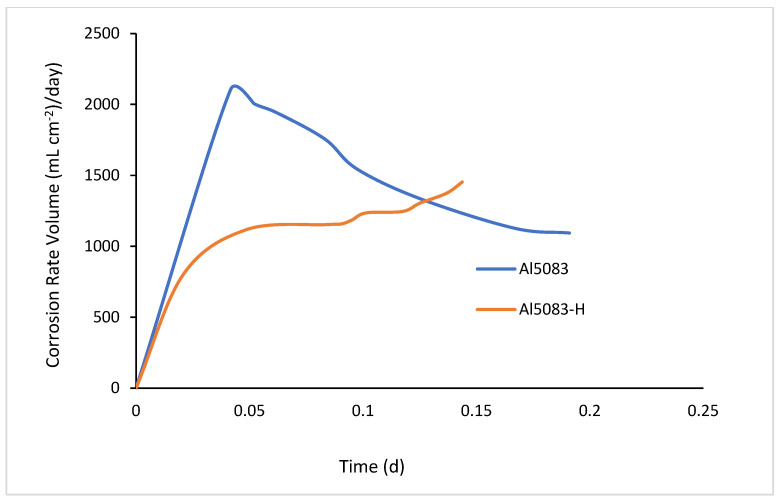
Hydrogen evolution corrosion rate volume results of Al5083 and homogenized (H) Al5083 in NaOH.

**Table 1 materials-17-03313-t001:** Chemical (XRF) composition (wt%) of Al 5083.

Sample	Mg	Si	S	Cl	Cr	Mn	Fe	Ni	Cu	Zn	Al
Al5083-H11	3.65	0.34	0.01	0.03	0.12	0.52	0.34	0.01	0.06	0.03	94.84

**Table 2 materials-17-03313-t002:** EDS findings (wt.%) of points 1 and 2 marked in Figure 4.

Spectrum	Mg	Al	Si	Cr	Mn	Fe
1	4.57	94.77	0.00	0.45	0.20	0.00
2	4.72	94.62	0.00	0.00	0.52	0.14

**Table 3 materials-17-03313-t003:** EDS findings (wt.%) of points 1 and 2 marked in Figure 5.

Spectrum	Mg	Al	Si	Cr	Mn	Fe
1	2.91	47.81	48.55	0.02	0.70	0.00
2	5.01	93.53	0.00	0.06	0.00	1.40

**Table 4 materials-17-03313-t004:** EDS findings (wt.%) of points 1–3 marked in Figure 10.

Spectrum	C	O	Mg	Al	Si	Mn	Fe
1	3.69	38.74	3.76	53.69	0.00	0.12	0.00
2	6.08	25.13	3.71	64.29	0.16	0.35	0.28
3	12.88	34.96	2.62	48.65	0.84	0.00	0.05

**Table 5 materials-17-03313-t005:** EDS findings (wt.%) of points 1–3 marked in Figure 11.

Spectrum	C	O	Mg	Al	Si	Mn	Fe
1	2.64	36.03	3.26	56.29	0.10	0.66	0.01
2	4.12	36.40	2.99	55.90	0.00	0.58	0.00
3	9.15	31.57	2.88	53.15	1.41	0.00	1.83

**Table 6 materials-17-03313-t006:** EDS findings (wt.%) of points 1–3 marked in Figure 12.

Spectrum	H	C	O	Na	Mg	Al	Si	Mn	Fe
1	3.44	3.28	8.67	0.53	4.05	76.59	0.00	0.94	2.51
2	11.19	3.34	27.59	0.46	2.34	53.27	0.00	0.66	1.17
3	3.64	7.05	14.16	0.86	3.17	66.14	0.42	1.28	3.29

**Table 7 materials-17-03313-t007:** EDS findings (wt.%) of points 1–3 marked in Figure 13.

Spectrum	H	C	O	Na	Mg	Al	Si	Mn	Fe
1	10.43	7.08	31.61	3.73	3.16	38.74	0.18	0.00	5.05
2	9.92	8.27	32.07	3.21	2.78	38.42	0.15	2.03	3.17
3	15.06	4.95	20.26	2.68	1.48	25.19	0.41	1.24	28.74

**Table 8 materials-17-03313-t008:** EDS findings (wt.%) of points 1–3 marked in Figure 20.

Spectrum	O	Na	Mg	Al	Si	Cl	Cr	Mn	Fe
1	26.14	2.20	3.69	62.48	3.62	1.44	0.43	0.00	0.00
2	14.45	0.60	3.46	72.26	3.68	0.29	0.31	0.59	4.36
3	10.93	0.69	2.56	68.74	3.55	0.21	0.95	3.17	9.19

**Table 9 materials-17-03313-t009:** EDS findings (wt.%) of points 1–3 marked in Figure 21.

Spectrum	O	Na	Mg	Al	Si	Cl	Cr	Mn	Fe
1	54.53	0.70	0.12	1.17	0.10	3.90	0.33	0.00	39.15
2	19.64	1.49	2.69	62.98	0.85	0.03	0.00	2.12	10.20
3	35.24	0.69	3.62	40.05	10.29	0.26	0.40	0.54	8.91

**Table 10 materials-17-03313-t010:** EDS findings (wt.%) of points 1–3 marked in Figure 24.

Spectrum	H	O	Na	Mg	Al	Si	Cr	Mn	Fe
1	43.11	19.75	0.17	0.55	5.71	0.00	3.22	4.54	22.96
2	40.68	0.66	0.13	3.06	54.69	0.00	0.00	0.79	0.00
3	62.58	9.02	2.23	4.85	16.71	4.29	0.00	0.29	0.02

**Table 11 materials-17-03313-t011:** EDS findings (wt.%) of points 1–3 marked in Figure 25.

Spectrum	H	O	Na	Mg	Al	Si	Cr	Mn	Fe
1	84.79	12.97	0.16	1.73	0.35	0.00	0.00	0.00	0.00
2	42.96	8.84	0.13	23.06	6.49	18.71	0.02	0.00	0.06
3	75.94	14.13	1.49	0.79	4.20	3.46	0.00	0.00	0.00

## Data Availability

The original contributions presented in the study are included in the article, further inquiries can be directed to the corresponding author.

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
