# Peer review of "Homogenization Heat Treatment for Enhancing Corrosion Resistance and Tribological Properties of the Al5083-H111 Alloy"

_materials, 2024, doi:10.3390/ma17133313_

Round 1

Reviewer 1 Report

Comments and Suggestions for Authors

The paper investigates the effects of homogenization heat treatment on the corrosion behavior and mechanical properties of Al5083-H111 alloy, providing valuable data and insights. However, there are several areas that could benefit from further improvement to enhance the overall quality and impact of the paper. The study provides valuable data and insights; however, there are several areas that could benefit from further improvement to enhance the overall quality and impact of the paper.

  1. Research Significance and Novelty: The research significance and the novelty of this work need further clarification. The introduction section would benefit from a more comprehensive review of the current state of research on the corrosion behavior of Al5083 alloy. Including more citations related to corrosion will help readers understand the actual progress in this area. I recommend expanding the literature review to highlight the advancements and existing gaps in the study of Al5083 alloy corrosion.

  2. Experimental Section: The experimental section could be strengthened with additional experiments to provide more robust theoretical support. For instance, the current study uses relatively short corrosion and hydrogen evolution times. Including long-term tests would be beneficial to evaluate the durability and reliability of the material under actual service conditions. This would provide a more comprehensive understanding of the alloy’s performance over extended periods.

  3. Mechanism Analysis: The mechanism analysis could be more detailed. The discussion on the corrosion mechanisms of Al5083 alloy in NaCl and NaOH solutions is relatively brief. Integrating microstructural analysis would provide deeper insight into the corrosion mechanisms. A more detailed examination of the microstructure and its relation to corrosion behavior would significantly enhance the mechanism analysis.

  4. Figures and Tables: The formatting of figures and tables in the manuscript appears unusual. For tables, I suggest using the three-line table format for clarity and consistency. Additionally, it would be more precise to report conclusions in the tables to two decimal places. This will enhance the readability and precision of the data presented.

Recommendations for Improvement:

  • Expand the introduction to include a detailed review of the literature on Al5083 alloy corrosion, highlighting recent studies and current knowledge gaps.
  • Include longer-term corrosion and hydrogen evolution tests to assess the material's long-term performance and provide stronger experimental support.
  • Conduct a more detailed microstructural analysis and discuss its implications for the corrosion mechanisms of Al5083 alloy in different environments.
  • Standardize the formatting of figures and tables, using the three-line table format and reporting data to two decimal places.
Comments on the Quality of English Language

The English language in the manuscript is adequate but could benefit from further refinement to improve clarity and readability. Consider having the manuscript reviewed by a native English speaker or a professional editing service.

Author Response

All comments of the reviewer have been taken seriously and the necessary corrections have been made.

Reviewer 2 Report

Comments and Suggestions for Authors

Manuscript ID: Materials-3071426-review-V1

The manuscript, titled " Investigation of Corrosion Behavior and Tribological Properties of Al5083-H111 Alloy" aims to enhance the corrosion behavior and tribological properties of Al5083-H111 alloy through homogenization heat treatment under different environmental conditions, including NaCl and NaOH for marine engineering, automotive manufacturing, and aerospace industries. Homogenization heat treatment refined the matrix structure of Al5083 alloy, increasing its hardness. Both treated and untreated samples showed better corrosion resistance and less weight loss in NaOH and NaCl environments, with NaOH causing higher weight loss and lower corrosion resistance compared to NaCl. Wear resistance in dry conditions was lower than in NaOH due to NaOH's lubricating effect and the enhanced corrosion resistance provided by oxide layers.

The following suggestions may improve the proposed manuscript:

Title:

·       The title should be modified to emphasize the research aim and novelty as follows: Homogenization Heat Treatment for Enhancing Corrosion Resistance and Tribological Properties of Al5083-H111 Alloy

In the abstract section:

·       State the name of the allow Al-Mg beside the code “Al5083-H111”

·       In lines 16-23, rewrite the following statements “It was observed that NaOH solution had lower corrosion resistance and higher weight loss compared to NaCl solution. …etc.”. The subject should be the alloy not the solution because the authors investigate the properties of Al503-H111 alloy in various conditions.

In the introduction section:

·       Provide detailed information on the effects of heat treatment on Al5083-H111 properties (microstructure, Hardness, corrosion resistance, COF, and wear resistance) from previous literature. Additionally, describe how various heat treatment parameters influence the properties of the alloy. Conclude by explaining why the following specific conditions were chosen for heat treatment: heating the sample at 500°C for 3 hours followed by air cooling.

·       The authors are encouraged to summarize the types of coatings utilized to enhance the corrosion and wear resistance of Al-Mg alloys. Among these, DLC and nanodiamond composite (NDC) hard coatings deposited via CAPD show promise. NDC represents a sustainable, wear-resistant, and durable carbon material.

In the methodology section:

1.     What are the benefits of hydrogen evolution test in this study?

In the Results section:

2.     XRD pattern of the homogenized alloy should be introduced and effects of heat treatment on microstructure should be pointed out.

3.     The two samples should be polished before imaging. Additionally, the resulting phases, grain size, porosity percentage, defect density, and residual stress should be studied. Techniques such as Electron Backscatter Diffraction (EBSD) and ImageJ software can be utilized for these analyses.

4.     Point out the correlation between the XRD and microstructure results.

5.     Fig. 2 and Fig.3 do not support your explanations.

6.     In section 3.3, State the causes behind hardness enhancement due to heat treatment.

7.     Add detailed information to Figures 16 to 19 to make it easier to distinguish and understand the differences between the various cases.

8.     In Section 3.5, Immersion Tests, the content is difficult to understand because it is presented as one long paragraph (lines 313 to 430). Break this section into smaller, more manageable paragraphs for better clarity.

Summery:

The manuscript presents valuable findings; however, it is recommended that the authors conduct a more in-depth investigation into the microstructural properties resulting from the heat treatments. This investigation should connect the microstructural findings directly with the mechanical, corrosion, and wear resistance observed. The decision regarding publication will depend on how well these aspects are addressed in response to the feedback provided.

Comments on the Quality of English Language

English evaluation:

The manuscript demonstrates a coherent structure but requires revisions to enhance clarity and readability. To improve, the authors should refine the writing flow by avoiding lengthy paragraphs.

Author Response

All comments have been considered and the corrections have been made.

Round 2

Reviewer 1 Report

Comments and Suggestions for Authors The authors have carefully addressed the review comments. Based on the overall quality of this manuscript, I think this paper can be accepted.

Reviewer 2 Report

Comments and Suggestions for Authors

The authors did significant improvements in the revised version. Thanks for considering the suggested comments.